# Palliative Care in the Delivery Room: Challenges and Recommendations

**DOI:** 10.3390/children10010015

**Published:** 2022-12-21

**Authors:** Lars Garten, Kerstin von der Hude

**Affiliations:** Department of Neonatology, Charité—Universitätsmedizin Berlin, 13353 Berlin, Germany

**Keywords:** neonates, life-limiting disease, dying, birth, symptom control, analgesia, bereaved parents

## Abstract

Palliative care in the delivery room is an interprofessional and interdisciplinary challenge addressing the dying newborn and parents as well as the caregivers. It differs in some significant aspects from palliative care in the neonatal intensive care unit. Clinical experience suggests that many details regarding this unique specialized palliative care environment are not well known, which may result in some degree of insecurity and emotional distress for health care providers. This article presents basic background information regarding the provision of palliative care to newborns within the delivery room. It offers orientation along with a preliminary set of practical recommendations regarding the following central issues: (i) the basic elements of perinatal palliative care, (ii) the range of non-pharmacological and pharmacological interventions available for infant symptom control near the end of life, (iii) meeting the personal psychological, emotional, and spiritual needs of the parents, and (iv) care and self-care for medical personnel.

## 1. Introduction

Despite advances in both prenatal and neonatal care, newborns represent the largest single group of deaths among children and adolescents, accounting for nearly 40% of cases [1,2]. Neonates die secondary to a wide range of causes, including complex congenital disorders, acute conditions specific to the neonatal period, or complications of extreme prematurity despite rapid advances in technology [3,4,5].

Easier access to highly technological antenatal care has significantly increased the number of prenatal diagnostic procedures [6,7,8]. In combination with an improvement in sensitivity and specificity, it is now more common for potentially life-limiting conditions (LLCs) that seem incompatible with long-term survival and/or carry the risk of the severe impairment of quality of life to be diagnosed during fetal life [9,10]. Families who discover through pre- or postnatal care that their babies may die before or shortly after birth are candidates for the specialized medical, psychosocial, spiritual, emotional, and practical support of perinatal palliative care (PnPC) [11,12,13].

Although the perinatal hospice concept was developed in the 1980s in the US [14], the first recommendations for the development of a PnPC pathway were only published in the early 2000s [15]. Since then, PnPC principles and guidelines have become well defined and standardized [16,17,18,19,20,21,22]. Today, many neonatal intensive care units (NICUs) have reached an inflection point where infant deaths following the limitation of life-sustaining treatments after the pre- or postnatal diagnosis of an LLC outnumber those following unsuccessful resuscitations.

A retrospective analysis of all mother–child pairs cared for between 2016–2020 by members of the PnPC program at the perinatal referral center at Charité Universitätsmedizin Berlin in Germany showed that two-thirds of all liveborns with a prenatally diagnosed LLC (e.g., trisomy 13/18, renal agenesia/dysplasia, complex congenital conditions or neurologic anomalies) died while still in the delivery room (DR) before admission into a neonatal ward [23].

In a prior study published in 2015, we demonstrated that the median documented survival time for newborns with LLC or extremely preterm infants at the limits of viability who were cared for under primary palliative care in the DR was 59 min and was not statistically different between specific subgroups [24]. Similar data on the duration of life in extremely preterm babies cared for under palliative care after birth were published by Durrmeyer et al. [25]. In their cohort of 73 extremely preterm neonates who died in the DR after decisions to withhold or withdraw life-sustaining treatments or after failed resuscitation, the median duration of life was 53 min.

Remarkably, within our 2015 study cohort of 113 patients, only 2 out of 113 patients received pharmacological treatment for symptom control [24]. This finding was consistent with formerly published data addressing the frequency of ‘comfort medication’ such as analgesic or sedative medication in dying neonates in the DR [5,26].

The above data already suggest that the palliative care of an infant in the DR differs in some significant aspects from other palliative care in other settings, including the NICU. Despite this, most physicians and caregivers currently lack access to either an evidence-based national database or to consensus-based recommendations appropriate for this special clinical situation.

The aim of this work is to provide clinically relevant background information and practical recommendations in the palliative care management of dying newborns in the DR. Specifically, it will address two issues: 1. What non-pharmacological or pharmacological options are available for symptom control to benefit newborn infants dying in the DR? 2. What general aspects of the psychological, emotional, and spiritual needs of the bereaved parents and families deserve particular attention? In addition, it will provide information about the basic elements of perinatal palliative care and support for medical personnel.

Aspects of this work have been orally presented at lectures, seminars, and workshops and a preliminary set of basic recommendations addressing palliative care in the DR have been previously published in German [27,28]. It is important to note that PnPC and symptom management extend beyond support at the end of life. The focus of this paper, however, is restricted to the provision of primary palliative care to neonates with LLCs dying in the DR following decisions to withhold or withdraw life-sustaining treatments.

## 2. Methods

Whenever available, background information and practical recommendations were developed based on the sparse evidence addressing primary palliative care in the DR. These evidentiary data were combined with (i) input from the “PaluTiN Group”, an interprofessional and interdisciplinary national expert panel that developed German recommendations for palliative care and grief counseling in peri- and neonatology [29], and (ii) personal clinical experience from our work at the PnPC team of the perinatal referral center at Charité Universitätsmedizin Berlin.

The perinatal referral center at Charité Universitätsmedizin Berlin consists of two NICUs, one intermediate care unit and one special care unit. At our institution, every year, approximately 30–40 neonates with LLCs are cared for under palliative care. Since 2016, our PnPC team has provided interdisciplinary and interprofessional prenatal palliative care consultations in collaboration with the prenatal diagnosticians, obstetricians, and midwives of our Department of Obstetrics as a part of family-centered fetal care. Our perinatal palliative care consultations are based on the principles of prenatal advanced care planning that have been published in detail previously [23,30].

The hospital-based PnPC team consists of:ten NICU nurses specialized in pediatric palliative care working in both tertiary NICUs or on our IMC unit;two perinatal social workers specialized in parental counseling and bereavement support;one neonatologist specialized in pediatric palliative care.

## 3. Recommendations

### 3.1. Fundamental Elements of Perinatal Palliative Care

Increasingly, holistic approaches and interdisciplinary concepts for palliative care in the DR are being offered to infants who are non-viable at birth. Interdisciplinary and interprofessional palliative care management, beginning with counseling at the time of prenatal diagnosis and extending to emotional support for the bereaved parents, has come to define the concept of perinatal palliative care [28,29]. Basic perinatal palliative care principles and guidelines have become well defined and standardized over the past two decades [16,17,18,22,29,31,32].

Most published recommendations for PnPC programs highlight the need to include specific core services [33]: a formal prenatal consultation; the development of a coordinated birth plan between the family, their obstetrician, and newborn caregivers; accessibility when needed to additional neonatal and pediatric specialists; comfort palliative care during the prenatal, birth, and postnatal periods; as well as psychosocial and spiritual support for families, siblings, and staff. Issues raised by bereaved parents offer insight into which aspects they see as being of particular importance. In one study, parents identified changes in communication partners and transfers of clinical responsibility among departments, the lack of coordination among the assigned medical, nursing, and psycho-social caregivers, as well as inconsistent communication and mixed messages as being the most stressful [34]. Parents also longed for recognition by the staff of their child as a person separate from their clinical status (“My baby is a person”). In this respect, parents reported being able to intuit a team member’s perception of their child’s worth by how the child was spoken about, whether as “your daughter” or “the baby”. In another study, the sense that they were being regarded by the treatment team as responsible, engaged, and thoughtful people, and were being treated as equal communication partners, were identified as important factors in their parental self-image and sense of autonomy [35]. Strong dissatisfaction with the birth situation or medical care was more likely to be expressed by respondents when doctors and nurses appeared not in accord with the parents’ decisions or expressed their own contrary moral judgments. Above all, the respondents expected to have an autonomous voice in the circumstances surrounding any interaction and handling of their child, particularly ones that influenced how much time would be available to them to be with their child after birth (e.g., waiving any possible resuscitating, spending time with their child while still alive or having sufficient time to say goodbye).

Planning for PnPC ought to simultaneously maintain a focus upon the medical and psychosocial support of the pregnant woman, taking into account pregnancy and childbirth along with anticipatory treatment planning for the child. A pregnant woman’s right to autonomy and self-determination has a fundamental priority during the prenatal care of both herself and her unborn child. In a multiple pregnancy, the medical needs of the unborn siblings, whether healthy or sick, must also be considered. PnPC requires a special diligence for these issues as well as for the broad range of treatment goals that need to be identified and weighed in their relative importance.


**KEY MESSAGES:**


Treatment goals that are essential for the successful provision of PnPC:*Assured continuity of interdisciplinary and interprofessional care for the parents and child;**Being open-minded and attuned to the parents’ needs when empathetically communicating information;**Thorough interdisciplinary and interprofessional planning for the child’s birth and postnatal care, planning that should serve the best interests of the child with consideration of the values, wishes, and needs of the parents;**Taking into consideration how the health concerns of the mother and possibly other siblings may be impacted by the prospective palliative care plan for the ill child;**Close coordination among all practitioners on the care team, including medical and nursing procedures;**Adequate symptom control for the newborn throughout the dying process;**The creation of valued memories for the bereaved parents;**Bereavement support for the parents and families.*

### 3.2. Non-Pharmacological Symptom Control

The idea of having to watch helplessly as one’s own child suffers “unnecessarily” due to pain or agitation in the dying process is unbearable for parents [36]. Only optimal symptom control enables parents to emotionally engage in their child’s dying process and subsequently say farewell. Consistent efforts to avoid pain and agitation plus effective treatment when they do occur are basic prerequisites for the successful end-of-life care of the child and an important consideration for the parents as they later cope with their grief.

Pain responses of newborns after acute procedures can be positively influenced by non-pharmacological measures. Interventions that include direct human physical contact have a visibly mitigating effect on observable pain responses [37,38,39,40,41]. Although lacking scientific proof of a specific link between the dying newborn’s physical contact with its parents and a reduction in its pain and stress, clinical experience and observational data cited in the studies above are consistent with the premise that such contact provides a meaningful measure of comfort as the child is dying. The direct physical contact between parent and child additionally supports the process of parent–child bonding and provides comfort to parents during their later grieving process [42,43]. This physical contact with the newborn during its postnatal palliative care should therefore ideally not be interrupted by interventions such as physical examination or monitoring.

For immediate postnatal end-of-life care, it is additionally recommended to avoid as far as possible any external stressors in the child’s environment such as bright lights, noise, or agitation. Similarly, any diagnostic not directly linked to relevant therapeutic consequences, including ECG monitor electrodes, oxygen saturation measurements, blood tests, and body temperature or blood pressure determinations, ought to be avoided.


**KEY MESSAGES:**

*Prevention is the most effective technique for successful pain and distress management for neonates receiving primary palliative care in the DR;*

*As a means of providing non-pharmacological pain and distress management for neonates with LLCs in end-of-life situations parents should be afforded the opportunity to provide uninterrupted comforting touch;*

*The effectiveness of non-pharmacological measures in postnatal palliative care in the DR can be further optimized by consistent avoidance of external distressing stimuli (e.g., bright lights, noise, or cold stress).*



### 3.3. Pharmacological Analgesia and Sedation

Experience drawn from numerous lectures, discussions, seminars, and workshops addressing the topic of “perinatal palliative care” has suggested to the authors that a significant degree of uncertainty prevails among nursing and medical professionals regarding pain assessment and the indications for pharmacological analgesia for neonates dying immediately after birth. For this reason, the issue of “pain and suffering in the dying phase” merits more detailed discussion. A fundamental question is often raised: does dying hurt? The first impulse among many nursing and medical staff working in neonatal intensive care units would be to answer with an emphatic “yes”. This perception derives in large part from common practice and scientific studies attesting to most neonates dying in a neonatal intensive care unit receiving concomitant potent analgesics—usually opiates—during the dying phase [5,26]. Yet, what evidence exists to support this use of analgesics? Or, expressed differently, what makes dying in a neonatal intensive care unit a painful process?

In many cases, newborns dying while in the intensive care unit suffer primarily from iatrogenic stressors and symptoms associated with various medical interventions such as invasive diagnostics, mechanical ventilation, or surgery. In contrast, a DR adequately prepared to deliver primary palliative care strenuously avoids such procedural pain whenever possible by intentionally refraining from any stressful invasive diagnostics or therapeutics.

A second common source of distress for neonates dying in the intensive care unit is disease-associated symptoms. Often, it is not the dying process itself that stresses the newborn but rather the noxious stimulation associated with such pathologies as necrotizing enterocolitis, capillary leak syndrome, or the increased intracranial pressure from intra-cerebral hemorrhage. Severe dyspnea, such as that seen with progressive or severe pulmonary insufficiency or heart failure, also commonly contributes to symptomatic distress. In the DR, however, the situation is usually different. The largest subgroup of newborns who die in the DR under primary palliative care is mostly extremely small premature babies at the limit of viability. Aside from their developmental immaturity, most commonly, these infants are not afflicted by a concurrent symptomatic disease process. What then causes these children to die? Experience shows that they die due to primary central apnea. Primary central apnea, however, is not accompanied by a subjective feeling of “air hunger”, because it is characterized by the absence of any respiratory drive. Accordingly, the infants do not show signs of “death throes” with tachypnea, dyspnea, or agitation. As a result of this postnatal central apnea, if not provided with respiratory support, these premature infants at the limit of viability will succumb to progressive hypercapnia and hypoxia. Together, these processes produce a natural form of sedation during the dying process.

An additional physiological feature may also play a role in diminishing the dying newborn’s ability to experience pain during the immediate postnatal period. It is known that the spontaneous birth process is invariably an extremely stressful and pain-associated process for both mother and child. The essential process of labor that produces spontaneous delivery is induced by the action of the hormone oxytocin, which is synthesized in the pituitary gland. In addition to promoting uterine contractions, oxytocin also functions as an endogenous, peripherally acting analgesic. The elevated peripartum serum levels observed during childbirth make the pain of labor and delivery bearable for the mother. The analgesic effect of oxytocin is mediated via the vasopressin 1 A receptor. Vasopressin [analog: Antidiuretic hormone (ADH) Adiuretin, or arginine vasopressin (AVP)] is also a nonapeptide hormone, differing from oxytocin only by two amino acids. Like oxytocin, vasopressin is also produced by nerve cells of the hypothalamus (nucleus supraopticus and nucleus paraventricularis), stored in the posterior lobe of the brain, and released into the blood as needed. During birth, the oxytocin level of the newborn changes very little in contrast to that of its mother. The newborn’s vasopressin level, however, increases by more than a hundredfold [44,45]. Likely, the high serum levels of vasopressin induced by the stress of birth produce physiological perinatal analgesia in newborns. It appears probable as well that this natural process affords an analgesic benefit during the immediate postnatal dying process covered by primary palliative management.

Only in rare cases, therefore, do the non-pharmacologic symptom control measures outlined above fail to provide adequate symptom control for the infant during an immediate postnatal dying process. When these exceptions occur, opioids represent the therapeutic option of first choice [46,47].

In choosing comfort medication for palliative care in the DR, two general aspects should always be taken into account: medication delivery should be non-invasive and convenient for weight-based dosing [48]. The use of intranasally administered fentanyl (note: off-label route) [49,50,51,52,53] meets both criteria. In addition, the intranasal administration of fentanyl avoids unnecessary disruption of the continuous physical contact between the dying infant and the parent that would occur during the process of establishing venous access. Most importantly, the anterior region of the nasal cavity affords a route for the rapid absorption and transfer of the drug into the systemic circulation. Unlike with oral administration, no first-pass effect occurs by this route due to bypassing the liver. In addition, the direct absorption of fentanyl into the cerebrospinal fluid may occur in the regio olfactoria, and from there into the brain. Most study data for intranasally applied opioids in the pediatric setting describe the use of fentanyl [54]. There is currently only one publication demonstrating that the use of intranasal fentanyl is safe and effective for symptom control in palliative care for newborns and infants up to 6 months old [52].

When administered by the nasal route, commercially available fentanyl injectable solution (0.1 mg/2 mL) can be used in single doses of 1–3 µg/kg [46]. We recommend:withdrawing 2 mL (100 µg) of fentanyl solution,adding 8 mL of normal saline (= final concentration of 10 µg/mL),administering 1–3 µg/kg = 0.1–0.3 mL/kg intranasally via a 1 mL tuberculin syringe.

For optimal absorption, the total targeted bolus can be divided into two doses, one dose given in each nostril. Significant pain reduction usually occurs as early as 5–10 min after intranasal application.

Newborn infants demonstrate highly variable opioid metabolism, necessitating that the fentanyl dose be individually titrated to achieve optimal symptom control without setting a predetermined maximum dose. There should be at least a 5–10 min interval between bolus administrations during the up-titration phase. It should be noted that non-soluble or poorly fat-soluble opioids such as morphine are not suitable for intranasal administration.

If the intranasal administration of fentanyl is not practicable, its delivery via the buccal route or the administration of morphine orally, buccally, or per rectum exist as alternatives. Note, however, that the administration of morphine requires a significantly longer time period to achieve a therapeutic effect compared to fentanyl when used as described above.

Pharmacological analgosedation should always be directed toward treating distressing symptoms and should not be given to accelerate the dying process. Gasping is a physiological process (‘reflex’) that occurs to varying degrees in every natural dying process. According to current knowledge, terminal gasping is not associated with distress and it cannot be prevented or treated by the application of opioids or sedatives [46].

For dosing regimens for fentanyl and morphine given via non-invasive routes, see Table 1.

A comparison of the different treatment approaches for common symptoms of distress between neonates dying in the immediate postnatal period and those dying later is summarized in Figure 1.


**KEY MESSAGES:**

*During the period immediately preceding a postnatal death, there exists a degree of physiologic analgosedation in vaginally born neonates and for extremely immature preterm infants attributable to elevated vasopressin levels, hypercapnia, and hypoxia;*

*The combination of physiologic analgosedation with the consistent use of non-pharmacological measures provides adequate symptom control in most cases;*

*If non-pharmacological measures fail to provide adequate symptom control, the use of intranasally administered fentanyl (single doses of 1–3 µg/kg) is the preferred recommendation;*

*Gasping is a physiological process (‘reflex’) that occurs to varying degrees in every natural dying process. According to current knowledge, terminal gasping is not associated with distress, and it cannot be prevented or treated by the application of opioids or sedatives.*



### 3.4. Supporting Bereaved Parents—Basic Aspects

The loss of a child immediately after birth confronts its parents both individually and as a couple with major emotional, psychological, and social challenges [13,55]. The premature death of their baby initiates for parents not only a grieving process for their child but also a denied opportunity to grow into their anticipated social role as parents, often accompanied by a devaluation of their self-esteem. These emotional issues lie at the core of the parental grieving process and are subject to being significantly influenced by psychosocial support provided to the parents [55]. Within any DR end-of-life situation, the nursing and medical staff bear a great responsibility for both the child and their parents. It is the task of the treatment team to help the parents focus their entire attention on the child without consideration of its brief lifespan and to recognize them as their own. This is especially important because many parents initially shy away from the idea of establishing contact with their dying newborn for fear that the resulting bond could intensify the pain of parting. Unresolved or ambivalent bonds, however, may promote a more difficult and emotionally complicated grieving process, often leading to depression and social isolation. Death with dignity in this setting also means allowing the manner and circumstances surrounding the dying process to reflect a respect for the cultural and religious background of the parents [13]. This process can be structured to include the creation of keepsakes (e.g., photographs, hand and footprints, or drawings) [56,57,58], which have been shown to exert a positive influence upon the bereaved parents’ subsequent grieving process [59,60].

In order to deepen their bond to the child as well as make the loss more tangible, parents need to be offered the opportunity to be in the accompanied presence of their deceased child—repeatedly if wished—until the funeral takes place. This might be arranged within the hospital by the provision of a separate “farewell room”, for example, or outside of the hospital, such as on the premises of the funeral home. Encounters with the child’s body help the bereaved parents discover the path to their personal mourning process. Consistent accompaniment from the beginning may also provide early insight into individual risks and resources that may attend to the parents. Another important point in this context is the psychological or pastoral support of parents [61].

Already in 2003, an epidemiological study from Denmark suggested how important this is. In this study by Li et al. [62], data from all children who died in Denmark between 1980–1996 were analyzed and compared to the subsequent mortality rates experienced by the bereaved parents. Within the total group of 12,072 children who had died, newborns accounted for 37%. The study revealed two important aspects:Bereaved mothers had a 40% increased risk of mortality during the first 3 years after the loss of their child, due to an increase in unnatural causes of death;The increase in maternal mortality was independent of the age of the deceased child. In terms of increased mortality, the death of a newborn appeared to weigh as heavily upon the mothers as did the loss of an older child who had been years longer a part of the family.

These data demonstrate that the death of a newborn is not a “passing event”, because it has a significant impact lasting well beyond the death of the child. The support of the bereaved parents of newborn babies should therefore not end with the death of the child, but rather be seen as a long-term obligation to provide needs-oriented support with the goal of helping the parents establish a secure everyday life [13,29,61].


**KEY MESSAGE:**

*The responsibility for primary palliative care in the DR does not end with the death of the child. It should include the continued accompaniment of bereaved parents outside of the hospital setting with needs-based support services to help them establish a secure daily life.*



For more details and practical advice about the bereavement support of parents and families within the context of perinatal palliative care: von der Hude, K and Garten, L. Psychosocial support within the context of perinatal palliative care: the “SORROWFUL” model.

### 3.5. Care and Self-Care for Medical Personnel

There is ample literature supporting the view that the end-of-life care of children can be highly stressful for medical staff, and situations likely to intensify this distress have been identified [63,64,65,66,67,68,69,70,71,72]. There are two main sources of distress for medical personnel: (1) patient quality care issues (e.g., caring for children with inadequately treated pain and dyspnea, “not having enough time for good care” because of staff shortages, etc.), and (2) moral and emotional stressors (e.g., as a consequence of missing precise treatment goals, being insecure how to communicate with parents losing their newborn child, etc.).

These insights suggest that improvement in institutional and organizational function, such as maintaining appropriate staffing levels with the provision of sufficient time for communication, should offer significant potential for reducing the most frequent sources of distress in DR end-of-life situations. As part of the education of medical personnel providing palliative care in the DR, information on psychological distress related to children’s deaths and bereavement care should be conveyed from an early stage. Educational training to improve communication skills with parents and other family members, optimally utilizing patient role players, may help strengthen staff self-confidence.

Additionally, periodic discussion forums within the institution may offer team members the opportunity to address issues and experiences in a constructive, respectful, and empathetic manner. Team members should be attentive to signs in themselves or their colleagues of emotional stress in coping with the care of dead or dying children and their families. Whenever the tolerance of one team member is exceeded, the responsibility should be assumed by another.

Fundamentally, any institution or facility offering PnPC to families assumes an obligation to provide the necessary personnel, space, and organizational resources to enable these requirements to be realized [29].

## 4. Conclusions

PnPC, with its associated supportive grief counseling, is a complex task, which differs in some ethical, medical, and psychosocial aspects from pediatric palliative care. Few studies examining the qualitative aspects of PnPC in terms of perceptions of care, decision-making and bereavement, however, specifically focus upon neonates with life-limiting conditions being provided with primary palliative care within the DR. Furthermore, currently, most physicians and caregivers cannot refer either to an evidence-based national database or to consensus-based recommendations appropriate for this highly specific clinical situation.

The recommendations presented here for the palliative care management of dying newborns in the DR derive from a greater preponderance of personal experience than of documented evidence. Nevertheless, the authors hope that this paper may offer practical help for health care providers. We believe that these recommendations have the potential to (i) improve the quality of care for newborn children with LLCs and their families, (ii) avoid unnecessary and burdensome measures, and (iii) focus upon goals that are valuable and meaningful to both child and family.

Self-evidently, these recommendations do not claim to be able to answer all questions. They should rather serve as an orientation, encouraging discussion and continuous development. Further recommendation updates will be required to continually improve the quality of primary palliative care for neonates with LLCs who die in the DR before being admitted to an NICU. The recommendations presented here relate to commonly encountered situations and the similar courses that they tend to follow but still leave room for adaptation and accommodation to the individual needs of affected children and parents.

## Figures and Tables

**Figure 1 children-10-00015-f001:**
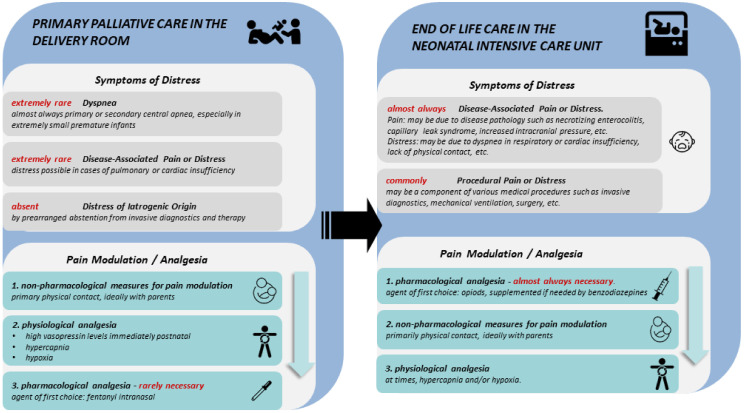
Pain and symptoms of distress during the process of dying—differences between primary palliative care in the delivery room and in end-of-life care the neonatal intensive care unit [adapted from Garten and von der Hude (2016) [27].

**Table 1 children-10-00015-t001:** Dosage by weight * for fentanyl and morphine given via non-invasive routes for symptom control for neonates under primary palliative care in the delivery room.

	Oral	Per Rectum	Intranasal	Buccal
**Fentanyl** Use injectable form [50 µg/mL], repeat administration every 5–10 min until optimal symptom control			1–3 µg/kg **	1–3 µg/kg
**Morphine** Use injectable form [2 mg/mL], repeat administration every 20–30 min until optimal symptom control	0.1–0.2 mg/kg	0.1–0.2 mg/kg		0.1–0.2 mg/kg

**Note**: * Titrate to effect with no maximum dose for opioids; ** preferred recommendation.

## Data Availability

Not applicable.

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
