# Peer review of "Palliative Care in the Delivery Room: Challenges and Recommendations"

_children, 2022, doi:10.3390/children10010015_

Round 1
Reviewer 1 Report
Thank you for asking me to review this paper on "Palliative care in the delivery room"; the topic is an important and, sadly, under-researched one. A review article about palliative care in the delivery room can contribute to the urgent conversation about support for babies/parents at time of the infant's death.
The authors state that 82.8% of deaths at Charité Universitätsmedizin happened in the context of prenatally prepared primary comfort care. The article covers support for parents during and after death, as well as pharmacological and non-pharmacological support for the infant at time of death. Given that a high number of cases take place in the context of "prenatally prepared primary comfort care", it becomes highly relevant to describe support that parents receive prenatally, when the incompatibility with life becomes apparent. I agree with the authors that "Interdisciplinary and -professional palliative care management, beginning with counseling at the time of prenatal diagnosis and extending to the emotional support for the bereaved parents, has come to define the concept of perinatal palliative care" I hope that the authors will consider reviewing strategies to support families before, and not only during and after birth. This gives families (not only parents but also siblings) the best chance to prepare and allows for the inter-disciplinary team to prepare a plan, alongside parents, to spend precious time with their baby, even if only for a few minutes after birth.
Reviewer 2 Report
The article presents a set of very important recommendations for the health teams working in these units. The implementation of these recommendations will allow a holistic approach to the newborn and his family, in the context of palliative care in the delivery room that dignifies the death of the newborn, as a person, the process of dying and the family's grieving process.
Reviewer 3 Report
Highly interesting!
Reviewer 4 Report
Lars Garten and Kerstin von der Hude presented in the manuscript titled '' Palliative care in the delivery room'' good information about the palliative care to newborns 10 within the delivery room and cares should be given to the parents as well. The manuscript is good and suitable for publication in Children after fullfill few comments:
1. The prevalence of death of newborns worldwide as statistic (table or graph) and preferably if related also to some specific reasons.
2. Statistic showing the types of pallitive care given to both newborns and parents worlwide (better if table or graph).
3. Pharmacological analgesia and sedation should be summarized in table showing the type of drugs, example, intervention case and possible reference.
Reviewer 5 Report
Thank you for an interesting manuscript on an important topic.
Although it is very interesting it needs a major revision.
A major revision is needed according to the following points:
1. The manuscript lacks a clear description of the methodology used. It looks like an opinion paper by experts from the field but a description of the methodology is lacking. Is it a kind of review or an opinion paper? Please state that clearly.
2. It is in some parts unclear if advices are based on own experiences, an expert group or the scientific literature.
3. Why is the retrospective examination based on data from 2000-2010 when the manuscript was written in 2022? Why are data from the last 12 years not included? Please include newer data material or explain the reason for your choice of the data in detail.
4. Are your NOTES more like key messages?
5. Please revise your manuscript using a sound methodology and state how your recommendations have been made and what the scientific evidence is.
6. Please add a proper conclusion based on a discussed on your data and scientific literature.
Good luck for a revision of your paper!
Round 2
Reviewer 5 Report
Thank you for the revision.
Now the reader can better understand your methods and suggestions.
I have no further comments.